# Reverse Engineering Orthognathic Surgery and Orthodontics in Individuals with Cleft Lip and/or Palate: A Case Report

**DOI:** 10.3390/bioengineering11080771

**Published:** 2024-07-31

**Authors:** Jaemin Ko, Mark M. Urata, Jeffrey A. Hammoudeh, Dennis-Duke Yamashita, Stephen L.-K. Yen

**Affiliations:** 1Craniofacial and Special Care Orthodontics, Division of Dentistry, Children’s Hospital Los Angeles, Los Angeles, CA 90027, USA; jako@chla.usc.edu; 2Division of Plastic and Maxillofacial Surgery, Children’s Hospital Los Angeles, Los Angeles, CA 90027, USA; murata@chla.usc.edu (M.M.U.); jhammoudeh@chla.usc.edu (J.A.H.); 3Division of Oral and Maxillofacial Surgery, Children’s Hospital Los Angeles, Los Angeles, CA 90027, USA; dyamashita@chla.usc.edu

**Keywords:** cleft lip and/or palate, virtual surgical planning, surgery-first approach, orthodontics, orthognathic surgery, three-dimensional planning

## Abstract

This case report presents a virtual treatment simulation of the orthodontic treatment and surgery-first orthognathic surgery employed to treat a patient with a repaired unilateral cleft lip and alveolus with Class III malocclusion and lower third facial asymmetry. The patient exhibited a negative overjet of 9 mm, a missing lower right second premolar, and a 5 mm gap between the upper right central and lateral incisors with midline discrepancy. The three-dimensional virtual planning began with virtual pre-surgical orthodontics, followed by the positioning of the facial bones and teeth in their ideal aesthetic and functional positions. The sequence of steps needed to achieve this outcome was then reverse-engineered and recorded using multiplatform Nemostudio software (Nemotec, Madrid, Spain), which facilitated both surgical and orthodontic planning. The treatment included a two-piece segmental maxillary osteotomy for dental space closure, a LeFort I maxillary advancement, and a mandibular setback with bilateral sagittal split osteotomy to correct the skeletal underbite and asymmetry. A novel approach was employed by pre-treating the patient for orthognathic surgeries at age 11, seven years prior to the surgery. This early phase of orthodontic treatment aligned the patient’s teeth and established the dental arch form. The positions of the teeth were maintained with retainers, eliminating the need for pre-surgical orthodontics later. This early phase of treatment significantly reduced the treatment time. The use of software to predict all the necessary steps for surgery and post-surgical orthodontic tooth movements made this approach possible. Multi-step virtual planning can be a powerful tool for analyzing complex craniofacial problems that require multidisciplinary care, such as cleft lip and/or palate.

## 1. Introduction

Cleft lip and/or palate (CLP) is one of the most common congenital orofacial defects in the United States, affecting one in 940 births [1]. This malformation derives from a disruption in the fusion of facial bones during craniofacial embryonic development. The cleft deformity poses unique challenges for clinicians as the CLP has several structural differences compared to the non-cleft condition. One major difference is the high prevalence of Class III malocclusions (skeletal underbites) in patients with CLP. The early surgeries employed to repair the CLP can cause scarring that restricts maxillary growth [2,3].

Several treatment modalities are available for addressing maxilla hypoplasia. Early maxillary protraction using reverse pull headgear during the early mixed dentition phase has been reported to be effective [4]. However, a frequent recurrence of Class III malocclusion is observed because maxillary protraction only targets the maxilla, allowing the mandible to outgrow the protracted maxilla during adolescence. Previous studies have reported success rates for early maxillary protraction ranging from 35% to 85% [5,6]. It was suggested that the long-term results of early protraction often provide only a temporary improvement, necessitating re-treatment during late adolescence [7]. Liou and associates developed an orthopedic technique combining alternate rapid maxillary expansion and constriction to efficiently advance the maxilla in CLP patients aged 9–12 years [8]. This technique has demonstrated an average maxillary advancement of 5.8 mm [8,9]. To address side effects such as open bite and frequent breakage of custom springs, this late protraction technique was further modified [10], and it has been reported to produce both skeletal and dental changes [11].

However, despite various non-surgical treatment options aimed at enhancing maxillary growth, many individuals continue to exhibit a short maxilla that cannot be adequately corrected with non-surgical orthodontic treatment alone. CLP is often regarded as a condition necessitating orthognathic surgery [12]. The need for orthognathic surgery has been reported in 25–60% of patients with unilateral cleft lip and palate and 65.1–69.5% of patients with bilateral cleft lip and palate, rates significantly higher than those reported in patients without a cleft lip or palate [13,14,15].

In addition to hypoplastic maxilla, individuals with CLP often have missing or malformed oral tissues. The teeth adjacent to the cleft area are frequently missing, malformed, or small [16,17,18]. Furthermore, the maxillary arch form is usually constricted and asymmetric in unilateral cleft cases [19,20]. When a canine is substituted for a missing lateral incisor, it may be difficult to obtain the ideal overjet, dental midlines, and intercuspation.

Here, we describe how virtual planning can be used to treat patients with cleft lip and palate. As an example, a case with unilateral cleft lip and alveolus is presented and the process of the virtual planning method is described. The planning was performed by combined surgical and orthodontic software, Nemostudio, (Nemotec, Madrid, Spain) that enables visualization of the final outcomes and predicts the changes performed during surgery [21,22,23].

A surgery-first approach was employed for the orthognathic surgery in this case. It has gained acceptance as a way to accelerate surgical–orthodontic treatment since Nagasaka and colleagues published the first case report of a systematic team approach between orthodontists and surgeons in 2009 [24]. However, many previous surgery-first studies often overlooked the importance of incorporating post-surgical occlusion and skeletal changes during the post-surgical orthodontic phase into the surgical planning [25,26]. In this report, the detailed process of determining transitional occlusion using reverse engineering is described.

It was not a surgery-first approach in the literal sense, as this patient had undergone orthodontic treatment at age 12, which could be considered an early pre-surgical orthodontic treatment. The outcome of this pre-treatment was maintained during adolescence so that the patient could proceed directly to surgery without any orthodontic appliances. This paradigm moves the pre-surgical orthodontic treatment to an earlier age when orthodontic treatment is normally carried out to promote improvements in dental esthetics at age 12.

## 2. Case Description

A 20-year-old male with a repaired unilateral cleft lip and alveolus presented to the craniofacial orthodontic clinic at the Children’s Hospital Los Angeles. The patient had been followed by our craniofacial team since infancy. He underwent lip repair surgery at 3 months of age. At age 10, his cleft space separating the cleft segments was bridged with a graft of bone that was harvested from the iliac crest. Following the eruption of the permanent dentition, orthodontic treatment was initiated to align the teeth into an arch form. This dental alignment became the pre-surgical phase of orthodontic treatment that coordinated the dentitions to fit properly for orthognathic surgery. 

At age 13, the brackets were removed to allow the patient to enter adolescent growth without braces. During adolescence, the patient’s mandible continued to grow anteriorly with minimal vertical changes, whereas the maxilla did not keep pace with the horizontal growth of the mandible. Due to the severe Class III skeletal discrepancy, our team decided that orthognathic surgery would be necessary after the completion of growth. The patient was followed by our team annually. Upon reaching 20 years of age, the patient elected to proceed with orthognathic surgery and returned to our orthodontic clinic for further treatment planning. Figure 1 shows his intraoral and extraoral photographs and lateral cephalometric and panoramic radiographs. His cephalometric measurements are shown in Table 1. He did not need 10 months of pre-surgical orthodontic treatment as his dental arches were aligned at age 12. In other words, his surgery at age 20 was already planned and prepared for when he was 12 years old.

## 3. Virtual Treatment Planning

The treatment plan was established based on the patient’s problem list. Given his age and his request to complete treatment before age 21, we decided to utilize the surgery-first approach for his orthognathic surgery to shorten the treatment time. Intraorally, the patient presented with a 5 mm gap between the upper right central and lateral incisors, and his upper right lateral incisor was smaller than the left one. To close the space, two options were considered (Table 2). The first option was to close the gap with orthodontic tooth movement. The second option was to close the gap through surgery by advancing the upper right posterior maxillary segment by 2 mm and rotating the left maxillary segment posteriorly and to the right by 2 mm (Figure 2A). 

With Option 1, the arch form can change following dental space closure. To accurately predict and calculate the necessary tooth movement and arch form change, the final ideal tooth positions were first virtually set up, and the necessary orthodontic tooth movement was calculated accordingly. In the setup phase, we also simulated the correct inclinations and alignment of the dentition.

The estimated duration of the orthodontic treatment phase was also calculated. Each type of orthodontic tooth movement has a different velocity of tooth movement. The software accounts for the distance a tooth needs to move and the time needed to achieve the normal rate of tooth movement in order to estimate the overall treatment time. After calculation, the time required for orthodontic space closure and decompensation was predicted to be 18 months.

Option 2 included surgical closure of the space by moving the segments together during the surgery, which called for a two-piece segmental osteotomy. This method required only minor tooth movement, such as leveling, alignment, and torque control for decompensation, as the space closure would be achieved surgically (Figure 2B). The software calculations predicted the orthodontic treatment would take 9 months. The planned procedure would approximate two maxillary segments by 4 mm after a two-piece segmental osteotomy cut at the cleft site. A 1 mm gap was intentionally left to widen the small lateral incisor using dental restoration materials, making it the same size as the contralateral incisor. 

After discussions about the treatment plans, the patient chose the second option to surgically close the dental spaces during the surgery to advance the maxilla. This option was predicted to also save 9 months of treatment time. Additionally, the patient’s cone beam computed tomography (CBCT) scans showed a thin and short alveolar bone in the area of the missing teeth, which was not ideal for orthodontic tooth movement (Figure 3).

### 3.1. Step 1: Virtual Orthodontic Setup

Virtual study models were generated from the initial intraoral scans and merged with the initial CBCT scans. The virtual orthodontic setup was achieved by segmenting each tooth and aligning the teeth into the arch form. The mandibular incisor inclination was increased so that it was perpendicular to the lower border of the mandible. Figure 4 shows the skeletal and dental positions after the virtual pre-surgical orthodontic treatment.

### 3.2. Step 2: Virtual Surgical Repositioning

The next step was surgical planning using the dental setup models completed in step 1 (Figure 5A). The surgical plan advanced the maxilla (LeFort I osteotomy) and set back the mandible (Bilateral sagittal split osteotomy) to correct the skeletal underbite. Simultaneously, the maxilla was divided into two segments to move the segments together to close the large interdental space in the cleft site. The maxilla with the corrected dentition was advanced in three-dimensional (3D) space to allow the facial surface of the maxillary central incisor to be roughly in a vertical line with the glabella, the anterior part of the forehead (Andrew’s FALL line) [27]. The vertical position of the central incisor was adjusted for incisal display and facial third proportions. Finally, the facial asymmetry, cant, pitch, and yaw were corrected to optimize facial aesthetics and occlusion (Figure 5A,B). Once the incisor and maxilla were set into the proper position in a manner similar to planning denture teeth in wax rims, the mandible and the mandibular dentition were brought into proper occlusion (Figure 5C). These positions were then fine-tuned to achieve the ideal occlusion, overjet, overbite, and occlusion (Figure 5D). This setup defined the target position of the jaws and teeth. The goal of this step was to generate the final skeletal and dental position, which would be used to reverse engineer the transitional occlusion for the surgical splint.

### 3.3. Step 3: Reverse Engineering the Transitional Occlusion

There are three occlusions that are needed when planning orthognathic surgery. The final occlusion represents the skeletal change and the bite at the end of treatment as defined in Step 2. In the surgery-first protocol, where there is no orthodontic treatment before surgery, the surgical changes to the maxillary and mandibular bones need to be recorded in the splint, which will be fitted onto the untreated initial dentition. Working backwards, the initial models without orthodontic treatment were swapped in for the final post-surgery orthodontically treated models planned in step 2 (Figure 6A). This is the transitional occlusion. The surgically advanced maxilla and mandibular setback positions are the same, but the models without any orthodontic treatment replaced the models that were orthodontically corrected (Step 1).

A proper transitional occlusion is crucial for successful post-surgical orthodontic treatment, as it reflects the skeletal movements during surgery and takes into account any orthodontic tooth movement that will be initiated after surgery. After the models were swapped, occlusal interferences caused by the malpositioned untreated teeth (Figure 6B) were addressed by a clockwise rotation of the mandible to open the bite and avoid any occlusal contacts (Figure 6C). The resulting open bite would later be closed with a counterclockwise mandibular rotation during post-surgical orthodontics. The overall process is summarized in Figure 7.

The patient underwent orthognathic surgery as planned. The surgical splint was maintained for 6 weeks to ensure transverse stability. Post-surgical orthodontic treatment commenced after splint removal. Minor crowding was corrected, and dentition was decompensated for during this phase (Figure 8). The retained primary lower right second molar was extracted during surgery, and the extraction space was maintained for future implant placement. Some alveolar bone loss was observed at the end of the treatment, which could have been prevented by the earlier removal of the tooth and space closure [28]. The orthodontic appliances were removed after 8 months of post-surgical orthodontics (Figure 9).

## 4. Discussion

The orthodontic treatment of patients with CLP occurs at different ages to prepare a patient for a bone graft at age 7–10, to align the teeth after permanent teeth erupt at age 11–14, and to prepare a patient for surgery after adolescent growth is complete at 17–20. In this case, the pre-surgical orthodontic treatment was completed when he was 13 as part of an earlier phase of orthodontic treatment to align the teeth. Instead of starting a second round of orthodontic treatment at age 18, which would normally take 10–14 months of pre-surgical orthodontics, the patient went directly to surgery because this pre-surgical treatment was completed at age 13 and maintained with retainers during adolescence. In this case, the overall treatment time for orthodontics and orthognathic surgery was reduced from 18–24 months to 10 months of treatment.

Treatment of CLP presents unique challenges, including the irregular shape of the cleft maxilla, the frequent absence or malformation of teeth, the need for dental substitutions and implants, and the variations in bone volume within the alveolar cleft site, as shown in Figure 10 [29,30]. In addition, the orthognathic surgery often required for CLP to correct maxillary hypoplasia has its own challenges due to anatomical variations and altered soft tissue conditions [31]. 

Multi-platform software integrates orthognathic surgery planning with orthodontic, implant, and prosthodontic planning to help clinicians visualize the final treatment outcome and develop treatment goals. Using the software is particularly beneficial when implementing the surgery-first approach. One key advantage of the surgery-first approach is higher patient satisfaction, as it eliminates the pre-surgical orthodontic phase and provides immediate enhancement of soft and hard tissue aesthetics [32,33]. However, some researchers have expressed concerns about the surgery-first approach, citing challenges in predicting post-surgical orthodontic outcomes and the potential for post-surgical instability [34].

It has been reported that the mandible tends to rotate in a counterclockwise direction and the chin point moves forward with the decrease of the mandibular angle after the surgery when using the surgery-first approach [35,36]. However, these changes after surgery are not due to surgical relapse but are attributed to occlusal changes during post-surgical orthodontic treatment [36,37]. In the surgery-first approach, traditional pre-surgical orthodontics for decompensation is performed post-surgically. Thus, planning for a surgery-first approach can be challenging as it requires a precise prediction of the post-surgical occlusion changes, which lead to alterations of the final skeletal position [38]. 

Overcoming these challenges is possible with reverse engineering in the planning. Reverse engineering starts by seeing the end and then stepping backwards step by step to design the steps needed to achieve the final outcome as shown in the case described [39]. There can be different paths or steps to achieve the same end, so the software allows clinicians to test different surgical and orthodontic options. 

Also, technological improvements in 3D virtual surgical planning have made this technology more accurate, accessible, and user-friendly. The software can enhance communication among the treatment team [40], which is important when multidisciplinary care is involved. 

In addition, 3D virtual planning offers a more comprehensive approach to visualize and measure maxillary and mandibular movements. Three-dimensional virtual planning can produce surgical guides and splints that precisely follow the surgical plan, allowing for better preparation and execution [41,42]. This method can minimize potential damage to the alveolar nerves by mapping out the position of the nerves in the post-surgical anatomy of the jaws [43,44]. Three-dimensional virtual planning has been shown to be an accurate and reproducible method for orthognathic treatment planning [45], reducing planning time and increasing the bone union rate [46,47,48].

Another advantage of 3D virtual planning software is the ability to 3D print surgical splints or guides used during surgery. It is possible to practice the surgery by 3D printing the entire mandible and maxilla and to make surgical guides that fit over bony anatomy. Custom surgical plates can be milled or cast based on the virtual surgical setup to create stiffer plates that resist surgical relapse. Clinicians may also use 3D-printed implants, such as malar implants, to change the bony contour of the underlying facial skeleton [49].

Despite the benefits of virtual treatment planning, there are limitations to its effectiveness. Precise virtual planning does not guarantee that actual treatment will exactly replicate the plan. Discrepancies can arise between the diagnostic virtual pre-surgical orthodontic setup and the actual pre-surgical orthodontic results [50]. Achieving the final outcome depends on the clinician’s ability to execute the plan [45]. Additionally, the movements simulated on a computer screen may not be biologically feasible due to scarring from earlier surgeries, which can restrict segmental movement during and after surgery [51,52]. This could explain why the patient described here had some remaining mandibular prognathic profile. To improve the transfer of predictions to actual treatment, future advancements could include the use of 3D-printed indirect bonding trays or prefabricated sequential clear aligners.

Several studies have compared conventional planning and virtual surgical planning in orthognathic surgery. While many studies reported improvements in the accuracy of planning and clinical outcomes [53,54], some studies highlighted drawbacks such as the cost of software and equipment, and the need for technical skills in manipulating the software [55]. 

During the planning process for the case described, we also encountered technical challenges. Adding virtual orthodontic planning to the surgical planning required additional effort and time. Several hours of training and practice were necessary to become proficient with the software, indicating a learning curve in the planning process. Additionally, predicting the direction and amount of decompensation required a thorough evaluation of the original compensation, which also added time to the planning process.

Choosing the right treatment option from these various modalities requires careful consideration of several factors. Individualized treatment plans are essential, as each cleft patient presents a unique combination of problems. Tailoring treatment options to address the specific needs of each patient ensures the chosen modalities are effective. Selecting the best treatment involves considering the patient’s unique circumstances, preferences, and the potential for successful outcomes.

In the case described above, several treatment options were considered to address the wide gap between the upper central and lateral incisors, as well as the deficient bone in the alveolar cleft site during the planning process. Virtual planning proved especially useful for planning surgery in patients with cleft lip and palate. As the patient had already undergone orthodontic treatment at age 13, the surgery-first approach was adopted to shorten the overall treatment time. The surgical and orthodontic software allowed for precise predictions of maxillary and mandibular repositioning, segmental widening, and post-surgical orthodontic tooth movement needed to finish the case to improve facial esthetics and occlusal function.

## Figures and Tables

**Figure 1 bioengineering-11-00771-f001:**
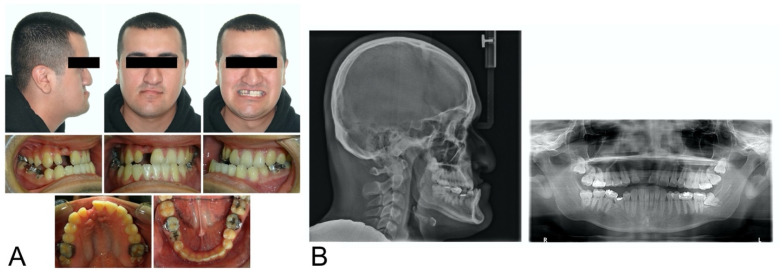
Initial records: (**A**) extraoral and intraoral photographs; (**B**) lateral cephalometric and panoramic radiographs.

**Figure 2 bioengineering-11-00771-f002:**
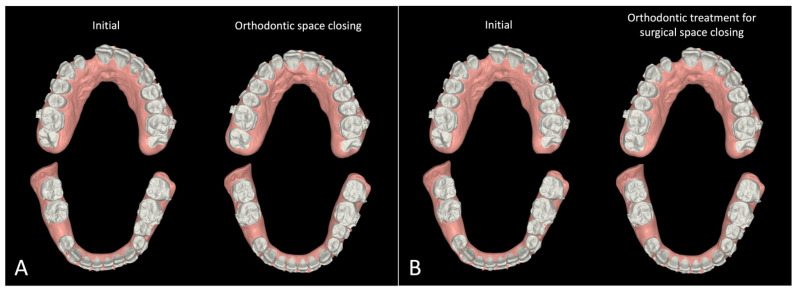
Simulation of treatment options for space closure: (**A**) orthodontic space closure; (**B**) orthodontic treatment for surgical space closure.

**Figure 3 bioengineering-11-00771-f003:**
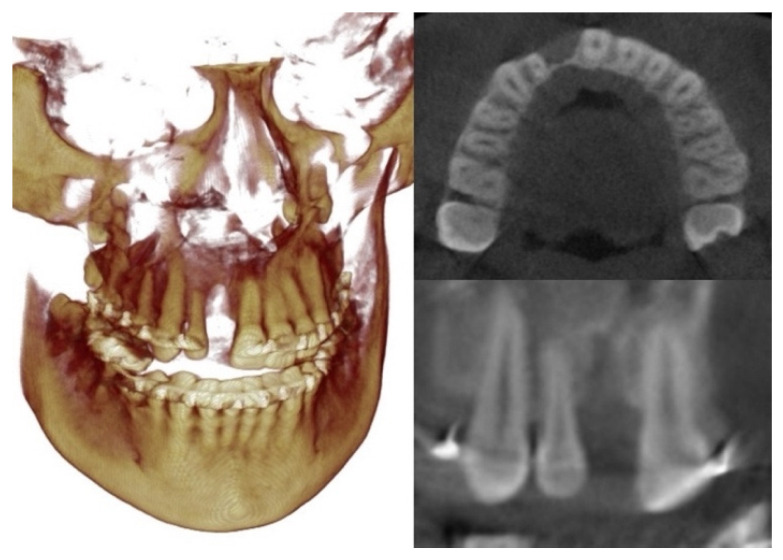
Evaluation of grafted alveolar cleft with CBCT.

**Figure 4 bioengineering-11-00771-f004:**
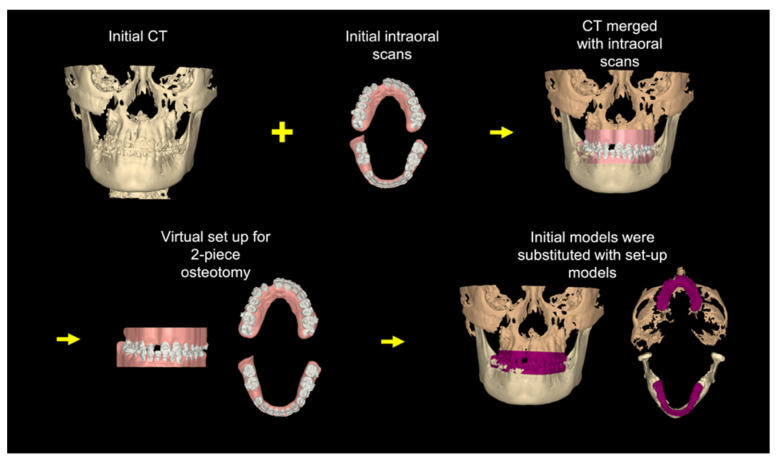
Step 1. Virtual orthodontic setup. Virtual setup models (purple) replaced the initial models (pink) in the initial CT scans.

**Figure 5 bioengineering-11-00771-f005:**
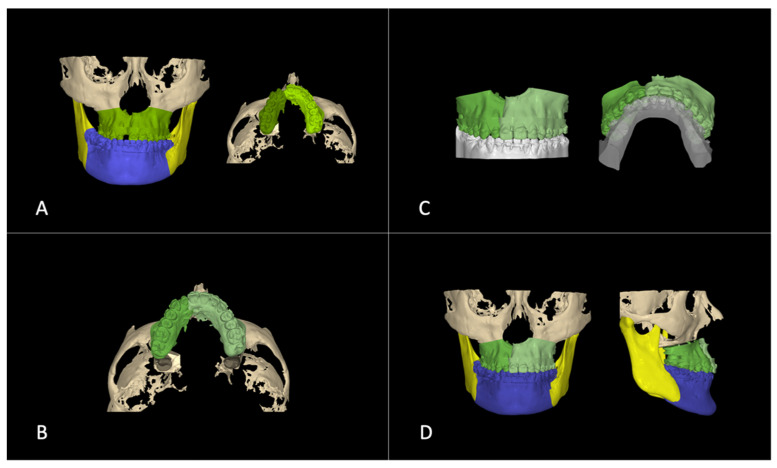
Step 2. Virtual surgical planning: (**A**) pre-surgical skeletal and dental positions after pre-surgical orthodontic setup, the two maxillary segments were defined by the osteotomy line; (**B**) the maxillary segments were approximated after 2-piece segmental osteotomy; (**C**) maxillary and mandibular models are fitted together to have ideal occlusion; (**D**) skeletal and dental positions after surgical repositioning.

**Figure 6 bioengineering-11-00771-f006:**
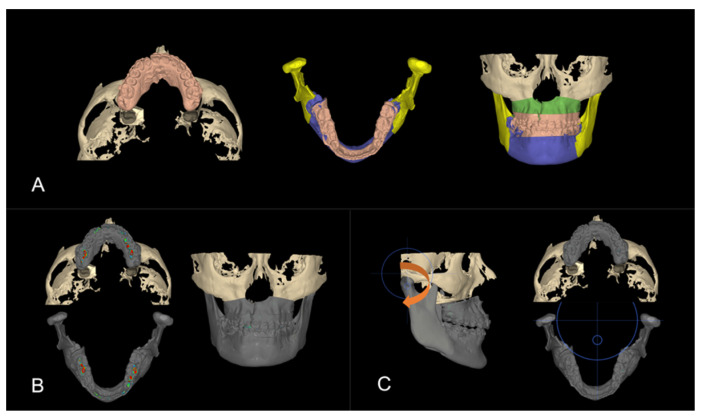
Step 3. Reverse engineering the transitional occlusion: (**A**) initial models (peach) substituted the final models from step 2 (Figure 5D); (**B**) occlusal inferences were visualized; (**C**) the mandible was rotated clockwise until all occlusal inferences were removed.

**Figure 7 bioengineering-11-00771-f007:**
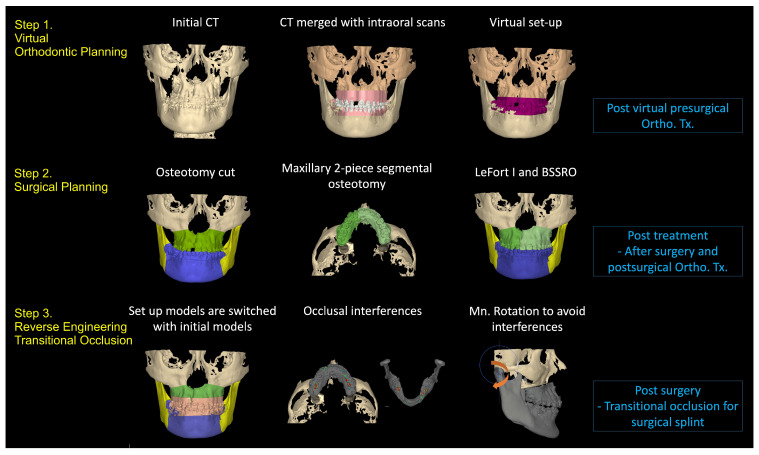
Summary of the process of virtual planning: step 1. Virtual orthodontic planning. The last image shows the skeletal and dental positions after the virtual presurgical orthodontic treatment; step 2. Surgical planning. The last image shows the final ideal skeletal and dental positions; step 3. Reverse engineering the transitional occlusion. The mandible is rotated to avoid occlusal interferences.

**Figure 8 bioengineering-11-00771-f008:**
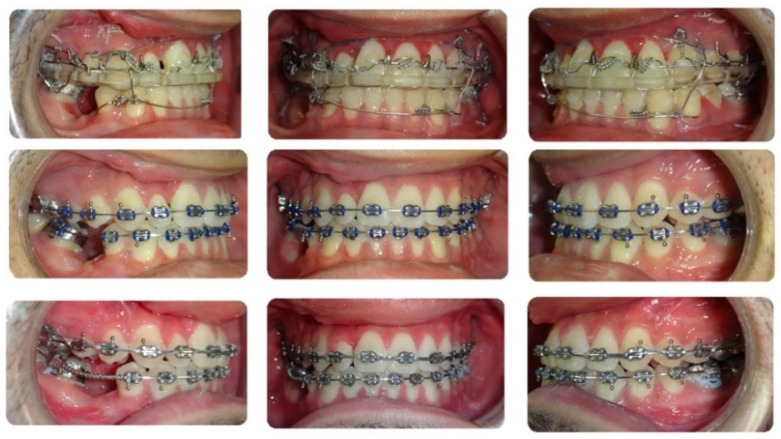
Progress intraoral photographs of the post-surgical orthodontics.

**Figure 9 bioengineering-11-00771-f009:**
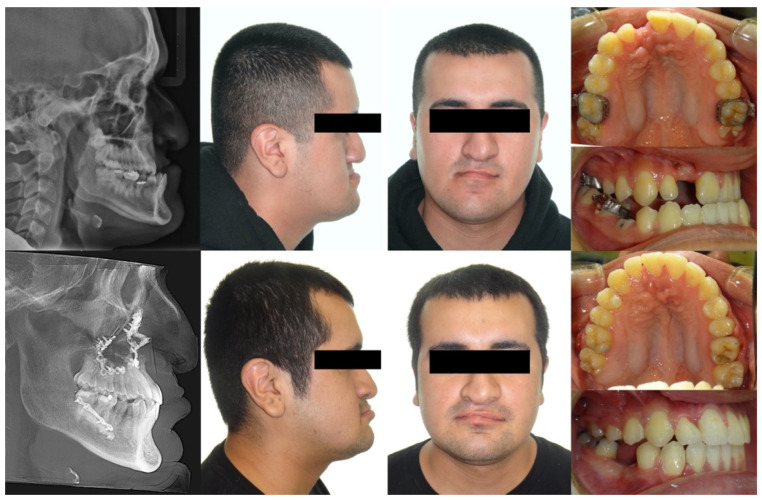
Initial and final photographs and radiographs.

**Figure 10 bioengineering-11-00771-f010:**
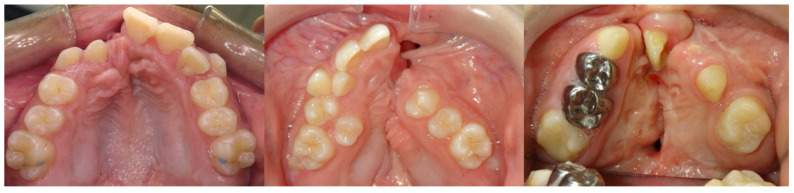
Typical intraoral presentation in cleft lip and/or palate. Maxillary arches are constricted due to prior surgical intervention and residual fistulas are visible due to unrepaired alveolar clefts.

**Table 1 bioengineering-11-00771-t001:** Cephalometric measurements.

Measurement	Normal	Pretreatment
Skeletal		
SNA (°)	82.0 ± 3.5	78
SNB (°)	80.9 ±3.4	87.7
ANB (°)	1.6 ± 1.5	−9.7
A-Na Perpendicular (mm)	1.1 ± 2.7	2.4
Pog-Na Perpendicular (mm)	−0.3 ± 3.8	30.5
Co-A (mm)	99.8 ± 6.0	79.8
Co-Gn (mm)	134.3 ± 6.8	132.6
Co-Gn—Co-A (mm)	34.5 ± 4.0	52.8
Wits (mm)	−1 ± 1.0	−17.4
SN-MP (°)	33.0 ± 6.0	27.1
Dental		
U1-SN (°)	103.1 ± 5.5	108.3
L1-MP (°)	95.0 ± 7.0	83.2
Soft tissue		
Upper lip to E-line (mm)	−4.0 ± 2.0	−14.7
Lower lip to E-line (mm)	−2.0 ± 2.0	−5.3

**Table 2 bioengineering-11-00771-t002:** Treatment options for space closure.

Option 1. Orthodontic Space Closure		Option 2. Surgical Space Closure	
Orthognathic surgery (surgery first)		Orthognathic surgery (surgery first)	
Maxilla	LeFort I osteotomy to advance the maxilla	Maxilla	LeFort I osteotomy to advance the maxilla
			2-piece segmental osteotomy to close the gap
Mandible	Sagittal split ramus osteotomy to setback the mandible and correct the facial asymmetry	Mandible	Sagittal split ramus osteotomy to setback the mandible and correct the facial asymmetry
Post-surgical orthodontics		Post-surgical orthodontics	
Space closure		Decompensation	
Decompensation		Finishing and Detailing	
Finishing and Detailing			

## Data Availability

Data are contained within the article.

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
