# Peer review of "Reverse Engineering Orthognathic Surgery and Orthodontics in Individuals with Cleft Lip and/or Palate: A Case Report"

_bioengineering, 2024, doi:10.3390/bioengineering11080771_

Round 1

Reviewer 1 Report

Comments and Suggestions for Authors

The following analysis provides an assessment of the study titled "Reverse Engineering Orthognathic Surgery and Orthodontics in Individuals with Cleft Lip and/or Palate," including its strengths, shortcomings, and suggestions for improvement.

Strengths:

a. Innovative Methodology:

The research presents a new method for orthognathic surgery and orthodontics that involves reverse engineering and virtual treatment planning. This novel approach has the potential to greatly enhance accuracy and results for those suffering from cleft lip and/or palate (CLP).

b. Detailed Case Description:

A complete chronology and exhaustive information about the patient's medical history are included in the comprehensive case description. Clinicians who want to duplicate or learn from the treatment strategy will find this level of detail useful.

c. Use of Advanced Technology:

The use of Nemotec software for three-dimensional virtual planning exemplifies how cutting-edge technology is used in healthcare settings. It highlights how these kinds of technologies can improve the way complicated surgical procedures are planned and carried out.

d. Clear Abstract and Introduction:

The abstract gives a summary of the study's goals, methods, and results, which helps viewers quickly grasp the most important points. The opening does a good job of setting the scene for the study by talking about how common and hard it is to treat CLP.

Weaknesses:

a. Absence of Comparative Analysis:

There is a lack of comparative analysis between the proposed methodology and traditional treatment methods. Comparing outcomes, costs, and patient satisfaction between different approaches would strengthen the study’s conclusions.

b. Technical Challenges and Limitations:

The paper doesn't talk about the technical problems and limits that came up during the process of virtual planning and reverse engineering. Taking these problems into account would give us a more balanced view.

In short, by addressing these issues will allow the study to evaluate reverse engineering orthognathic surgery and orthodontics in cleft lip and palate patients more thoroughly and fairly.

Author Response

Comment 1: There is a lack of comparative analysis between the proposed methodology and traditional treatment methods. Comparing outcomes, costs, and patient satisfaction between different approaches would strengthen the study’s conclusions.

Response 1: Thank you for your time to review our paper, and thank you for pointing this out. We have included comparative analysis between the method we used and conventional planning method in the discussion section. The benefits and drawback of the methods were discussed as you suggested (line 286-301)

Comment 2: The paper doesn't talk about the technical problems and limits that came up during the process of virtual planning and reverse engineering. Taking these problems into account would give us a more balanced view.

Response 2: We appreciate your constructive feedback. We added limitations of the virtual treatment planning and described technical challenges we encountered using the method. ( line 302-307)

Reviewer 2 Report

Comments and Suggestions for Authors

Interesting paper

I have some suggestions to improve the presentation quality of the paper 

Title; Add that this is case report

I am not sure ‘reverse engineering ‘is the correct term; use ‘treatment simulation’ change this in the text as well

Abstract: start by presenting the case ; this case present treatment simulation of combined orthodontic and orthognathic surgery (surgery first protocol)to treat a male unilateral cleft patient with Class III malocclusion on a Class III skeletal base with lower third facial asymmetry complicated by reverse over-jet of ,,, missing lower second premolar, centre line discrepancy, and small upper right lateral incisor, patient has lip repair at age,, and bone graft at age,,,,

Add the city, company name for the software use and follow up period

Introduction, very short and deficient, line 43, also add that one of the indications for orthognathic surgery is having cleft or Craniofacial anomalies and it is well starblished in the new index of IOFTN(J Orthod. 2014 Jun;41(2):77-83.;Cleft Palate Craniofac J. 2023doi: 10.1177/10556656231216833. )

case summary, add macnamara analysis for position of maxilla and mandible

table 2, add in each option what was the objective of Lefort I and BSSO ( advance the maxilla and setback the mandible and correct the facial asymmetry)

Discussion, disaccharide the limitations of this approach in more detail

although surgery first protocol reduced treatment time but limited the decompensation and outcome somehow as after the surgery the patient profile is still with a slightly prognathic mandible , a better treatment option would have been decompensation and more forward movement of maxilla , discuss this in detail

ALSO INCLUDE OTHER TREATMENT OPTIONS SUCH as LATE MAXILLARY PROTRACTION FOR THIS CASE THAT COULD have ADVANCEd THE MAXILLA MORE (Cleft Palate Craniofac J. 2014 Jan;51(1):e1-e10)

early space closure and removal of retained lower right E also could have prevented the bone volume deficiency in the area (J Oral Implantol. 2012 Dec;38(6):779-91.)

Comments on the Quality of English Language

Paper needs revision

Author Response

Comment 1: Title; Add that this is case report

Response 1: We appreciate your time reviewing our paper. The title was changed to "Reverse engineering orthognathic surgery and orthodontics in individuals with cleft lip and/or palate: A Case Report" 

Comment 2: I am not sure ‘reverse engineering ‘is the correct term; use ‘treatment simulation’ change this in the text as well

Response 2: Thank you for your insightful feedback regarding the use of the term 'reverse engineering.' However, we believe that this term is essential to highlight the distinction from previously reported surgery-first approach planning methods. Several studies reported that the mandible tends to rotate counterclockwise and the chin point moves forward with a decrease in the mandibular angle after surgery when using the surgery-first approach. Studies suggested that these skeletal changes are attributed to occlusal changes during postsurgical orthodontic treatment rather than surgical relapse. It means that, prediction of occlusal changes that is happening during the decompensation process in postsurgical orthodontics is crucial as post surgical skeletal position changes will be induced by this. Unfortunately, many previous studies on the surgery-first approach have overlooked the importance of integrating post-surgical occlusion and skeletal changes into surgical planning. In this case report, the dentition that underwent virtual presurgical orthodontics was used to plan the orthognathic surgery (final dental and skeletal positions), and then the transitional occlusion (the temporary malocclusion right after the surgery) was determined by reverse engineering from the final dental and skeletal positions. Setting the right transitional occlusion is the key to success in surgery-first approach as this is the position for the final splint which guides surgical repositioning. It is for this reason that we want to keep the term "reverse engineering".  We realized that this point was not well articulated in the original manuscript. Therefore, we have added this detailed explanation with references to the introduction and discussion sections to clarify our approach (line 74-81, 244-255, 257-258). Thank you again for your valuable comments.

Comment 3: Abstract: start by presenting the case ; this case present treatment simulation of combined orthodontic and orthognathic surgery (surgery first protocol)to treat a male unilateral cleft patient with Class III malocclusion on a Class III skeletal base with lower third facial asymmetry complicated by reverse over-jet of ,,, missing lower second premolar, centre line discrepancy, and small upper right lateral incisor, patient has lip repair at age,, and bone graft at age,,,,

Response 3: Thank you for your feedback. I appreciate your suggestion to start the abstract by describing the case, which I have implemented in the revised version on line 12. 

Comment 4: add the city, company name for the software use and follow up period 

Response 4: It was added to the manuscript as suggested (line 19 and 71).

Comment 5: Introduction, very short and deficient, line 43, also add that one of the indications for orthognathic surgery is having cleft or Craniofacial anomalies and it is well starblished in the new index of IOFTN(J Orthod. 2014 Jun;41(2):77-83.;Cleft Palate Craniofac J. 2023doi: 10.1177/10556656231216833. )

Response 5: Thank you for your valuable feedback. We have expanded the introduction section to provide a more comprehensive overview of various modalities to correct maxillary deficiency, the necessity of orthognathic surgery in cleft patients, and a detailed description of the surgery-first approach and reverse engineering. Also, in response to your suggestions, we have also cited the two papers you recommended. These additions can be found on lines 42-55, 58-59, and 74-81.

Comment 6: case summary, add macnamara analysis for position of maxilla and mandible

Response 6: Macnamara analysis was added in table 1. 

Comment 7:  table 2, add in each option what was the objective of Lefort I and BSSO ( advance the maxilla and setback the mandible and correct the facial asymmetry)

Response 7: Thank you for your recommendation. We have added the objectives of LeFort I and BSSRO to Table 2 as suggested.

Comment 8: Discussion, disaccharide the limitations of this approach in more detail

Response 8: Thank you for your valuable feedback. We have provided a more detailed discussion of the limitations and technical challenges of the approach in the discussion section (lines 286-307).

Comment 9: although surgery first protocol reduced treatment time but limited the decompensation and outcome somehow as after the surgery the patient profile is still with a slightly prognathic mandible , a better treatment option would have been decompensation and more forward movement of maxilla , discuss this in detail

Response 9: Thank you for your valuable insight. We agree that greater forward movement of the maxilla could have resulted in a better treatment outcome. This could be due to the limitations of the virtual planning and we have addressed this issue in detail on lines 286-296. 

Comment 10:  ALSO INCLUDE OTHER TREATMENT OPTIONS SUCH as LATE MAXILLARY PROTRACTION FOR THIS CASE THAT COULD have ADVANCEd THE MAXILLA MORE (Cleft Palate Craniofac J. 2014 Jan;51(1):e1-e10)

Response 10: Thank you for your suggestion. We have further elaborated on the different treatment modalities including late protraction in the introduction section and have cited the recommended paper accordingly. These changes can be found on lines 49-55.

Comment 11: early space closure and removal of retained lower right E also could have prevented the bone volume deficiency in the area (J Oral Implantol. 2012 Dec;38(6):779-91.)

Response 11: Your suggestion has been added to the manuscript and cited. (line 216-218)

Reviewer 3 Report

Comments and Suggestions for Authors

very interesting and new field

do you use this software for something else?

you should

Author Response

Thank you for taking your time to review our paper. 

Round 2

Reviewer 2 Report

Comments and Suggestions for Authors

Thank you for the revisions 

Comments on the Quality of English Language

Minor copy editing maybe needed